# The Anti-Inflammatory Effect of Carrageenan/Echinochrom Complex at Experimental Endotoxemia

**DOI:** 10.3390/ijms231911702

**Published:** 2022-10-03

**Authors:** Irina M. Yermak, Anna O. Kravchenko, Eleonora I. Khasina, Ekaterina S. Menchinskaya, Evgeny A. Pislyagin, Ekaterina V. Sokolova, Galina N. Likhatskaya, Dmitry L. Aminin

**Affiliations:** 1G.B. Elyakov Pacific Institute of Bioorganic Chemistry, Far Eastern Branch, Russian Academy of Sciences, 100 Let Vladivostoku Prosp., 159, 690022 Vladivostok, Russia; 2Federal Scientific Center of the East Asia Terrestrial Biodiversity, Far Eastern Branch, Russian Academy of Sciences, 100 Let Vladivostoku Prosp., 159, 690022 Vladivostok, Russia

**Keywords:** lipopolisaccharide, endotoxinemia, carrageenan, echinochrome, complex, ROS, NO, cytokines, molecular docking

## Abstract

The anti-inflammatory effects of the CRG/Ech complex in LPS-induced endotoxemia were investigated in vivo in mice and in vitro in LPS-stimulated RAW 264.7 cells and peritoneal macrophages. The results indicated that the CRG/Ech complex suppressed the LPS-induced inflammatory response by reducing the production of ROS and NO in the macrophages. Furthermore, the in vivo experiment indicated that the CRG/Ech complex minimized disorders of the physiological and metabolic processes in mice subjected to LPS intoxication and reduced the levels of proinflammatory cytokines in the mouse serum. The preventive administration of the CRG/Ech complex to mice prevented endotoxin-induced damage in the mouse model of endotoxemia, increased the mice’s resistance to LPS, and prevented increases in the levels of proinflammatory cytokines (TNFα). In this work, we showed by the molecular docking that Ech interacted with carrageenan, and that H-donor and H-acceptor bonds are involved in the formation of the complex.

## 1. Introduction

Lipopolysaccharide (LPS) is the major component of the outer membrane of the Gram-negative bacteria. This endotoxin is composed of three modules: a highly variable O-antigen comprising repeating oligosaccharide units, a core oligosaccharide, and lipid A [1]. When released from bacteria in vivo or administered in isolated forms, endotoxins exert both physiological and powerful pathophysiological effects in higher organisms and, thus, represent important virulence factors of Gram-negative bacteria. LPS is the primary target for antibacterial drugs and components of the immune system of the host. Monocytes, macrophages and other types of immune cells can be activated by LPS, inducing the production of TNF-α, IL-6, IL-8 and free radicals, such as reactive oxygen species [2]. Although the immune status of the host, including the inflammatory process, is known to play an important role in resistance to various infections, the overexpression of these proinflammatory mediators may result in fever, coagulopathy, endothelial dysfunction, vascular instability, apoptosis, and multiorgan failure or septic shock. Circulating LPS is bioactive in vivo and correlates with measures of innate and adaptive immune activation [3]. Rich sources of LPS reside in the upper respiratory and GI tracts especially in the mouth and colon [4]. Compromised intestinal integrity and increased intestinal permeability favor LPS translocation from the intestinal lumen to the bloodstream, causing metabolic endotoxemia [5,6]. The condition of “metabolic endotoxemia” is caused by LPS and characterized by low-level inflammation, insulin resistance, and increased cardiovascular risk, as well as atherosclerosis, type II diabetes mellitus and Parkinson’s disease, as described in detail in the reviews [5,6,7,8,9].

The removal and detoxification of Gram-negative bacterial LPSs remains relevant. With the emergence of bacterial pathogens resistant to the effects of the main classes of antibiotics, the application of natural substances is currently of interest. So, Li et al. showed that resveratrol, due to its anti-inflammatory ability, can be an effective therapeutic agent for the treatment of depressive behavior caused by LPS [10]. The efficacy of polyphenols against endotoxins has been also reported in animal studies [11]. In an LPS-induced endotoxemia mouse model, it was shown that hirsutanol A (HA), isolated from the red alga-derived marine fungus *Chondrostereum* sp., improved endotoxemia-induced acute sickness behavior, including acute motor deficits and anxiety-like behavior [12].

Marine polysaccharides, as safe and natural polymers against bacterial diarrhea, have been considered as an alternative to antibiotics and are given preference due to their broad range of physiological activities and biocompatibility [13,14,15,16]. Recently, it was shown that sulfated polysaccharide from seaweeds (SFPS) suppressed the LPS-induced inflammatory response by reducing the production of NO and PGE2; inhibiting the expression of iNOS and COX-2; and decreasing the levels of proinflammatory cytokines in RAW 264.7 cells. Furthermore, in vivo experiments indicated that SFPS significantly improved the survival rate, suppressed heartbeat disorders, and reduced the levels of ROS, cell death, and NO in LPS-treated zebrafish [17]. It was also demonstrated that fucoidan from brown algae prevented endotoxin-induced damage in a mouse model of endotoxemia and increased the mice’s resistance to LPS [18]. Furthermore, β-glucans and β-glucan-containing compounds can act on several immune receptors and activate groups of immune cells, such as macrophages, neutrophils, monocytes, natural killer cells, and dendritic cells, and therefore modulate both innate and adaptive response mechanisms [19]. It was shown that the sulfated galactans from *Gracilaria ornate* exhibit antibacterial activity against two Gram-positive and five Gram-negative bacteria [20]. The potential application of sulfated polysaccharides from red seaweeds, namely carrageenans (CRGs), as protectors against the S and R forms of *P. mirabilis* and *B. cepacia* (1) LPS was demonstrated by our group [21]. Previously, we also revealed the potential application of CRGs as immunomodulators for the treatment of Gram-negative infections associated with the accumulation of endotoxins in an organism and we showed the therapeutic action of food-additive CRGs in complex therapy in patients with intestinal infections of *Salmonnela* etiology [22]. In recent years, CRGs have been widely present in the pharmaceutical sciences, with an increasing interest in integrating them as materials for the incorporation of bioactive agents [23]. Due to their strong ability to adsorb water, CRGs can improve drug dissolution and thus increase the oral bioavailability of poorly water-soluble drugs [24]. We have previously shown that CRG improves the water solubility of the water-insoluble sea urchin pigment echinochrome (Ech) and reduces its oxidative degradation [25]. The spectrum of the biological protective effects of Ech is very diverse, primarily due to its ability to neutralize the negative effects of free radicals. The therapeutic effect of Ech in endotoxin-induced uveitis was examined in an animal model [26].

Ech is the active substance (P N002362/01) of the drug Histochrome, which is registered in the Russian Federation as a solution for injections and is used for the treatment of ocular diseases [27]. However, Histochrome is easily oxidized due to the oxygen content in the air. The enclosing of Ech in the CRG can be useful for both the oral and topical administration of Ech, with a possible enhancement in its pharmacological action, taking into account the LPS-neutralizing effect of CRG.

The aim of this work was to investigate the effect of Ech in complex with CRG on LPS-induced endotoxemia in mice.

## 2. Results

### 2.1. Carrageenan and Echinochrome

CRGs are sulfated linear galactans, whose basic structural units are disaccharide, carrabiose, consisting of alternating β-1,3-linked and α-1,4-linked galactose residues. Variation on this basic structure results from the content of 3,6-anhydrogalactose, location, and number of sulfate groups [14,23]. Polysaccharide was extracted from the red seaweed *C. armatus* and separated using 4% KCl fractions. The structure of KCl-insoluble was studied by of ^13^C NMR and FT-IR spectroscopy and the obtained spectra were compared with the spectra of polysaccharides isolated by us earlier from these species of algae. The identity of the spectra indicated that the KCl-insoluble fraction from *C. armatus* was κ-CRG [28].

The standardized substance Ech (pentahydroxyethyl-1,4-naphthoquinone) was used as ethanolic solution. Ech is known to be water insoluble. We used an ethanolic solution of Ech in the concentration of 10 mg/mL as a stock solution. Water solutions of CRG/Ech were obtained as described in the methods.

The chemical structures of the Ech and disaccharide repeating units of the CRGs are listed in the Appendix A.

### 2.2. Molecular Docking

Previously, we showed by spectroscopy that CRG interacts with Ech [25]. In this work, molecular docking was used to prove the formation of the complex and to visualize and identify possible binding sites. We used the crystal structure of Ech (CCDC ID NERLUS) and the GRAMM program for docking with κ-CRG fragment. To predict the structure of a complex, the GRAMM program only requires the atomic coordinates of the two molecules (no information about the binding sites is needed). The molecular docking of Ech and a fragment of the κ-CRG double helix showed that Ech interacts with CRG and bound water molecules through the formation of H-donor and H-acceptor bonds (Figure 1 and Figure 2). The total value of the binding energy, −16.8 kcal/mol, confirms the possibility of complex formation. The 3-linked β-D-galactopyranose 4-sulfate residues of both CRG chains are involved in the formation of the complex.

### 2.3. The Cytotoxic Activity

Experiments to evaluate the cytotoxic activity of the CRG/Ech complex on RAW 264.7 cells showed that the complex does not suppress the viability of macrophages in the studied CRG/Ech concentration range (1.75–112 μg/mL CRG and 0.18–12.5 μg/mL Ech). It was also shown that alone Ech at the maximum studied concentration of 25 μg/mL, reduces cell viability by no more than 10% compared to the control (Appendix A).

### 2.4. Effect of Ech and CRG/Ech Complex on Levels of ROS and NO Induced by LPS

In the current study, we examined the influence of Ech alone and the CRG/Ech complex on the in vitro production of ROS and NO induced by LPS in the murine macrophage cell line RAW 264.7 and peritoneal macrophages at 1 and 24 h. ROS and NO synthesis in the presence of Ech alone or the CRG/Ech complex was determined at the ratios of the initial components (10:1), where the Ech concentration varied from 0.75 to 12.5 μg/mL, respectively, and the concentration of CRGs, from 7 to 112 μg/mL. As our results showed, pretreatment of cells with the studied samples reduces the synthesis of ROS and NO in the cells induced by LPS during incubation for 1 h. In all experiments, a dose-dependent effect of the influence of Ech and CRG/Ech on the ROS and NO production was found. However, it should be noted that in the case of peritoneal macrophages at high concentrations of both Ech and its complex with CRG, the most decreases of ROS level induced by LPS was observed for 1 h (Figure 3C).

According to our results, the ROS and NO production induced by LPS in the cell line RAW 264.7 as a positive control was approximately 3 and 1.8 times greater than that in the negative control. Ech and the CRG/Ech complex, after incubation with cells for 24 h, decreased the ROS formation induced by LPS by approximately 50–60%, regardless of the Ech concentration (Figure 4A). However, Ech alone showed a slightly greater inhibitory effect on the ROS synthesis induced by LPS, compared to the CRG/Ech complex, at high concentrations (12.5 and 6.2 μg/mL). A significant dose-dependent effect of inducing a reduction in NO levels caused by the presence of LPS was found when RAW cells were incubated with the CRG/Ech complex. The greatest inhibitory effect was observed at a complex concentration of 7 + 0.75 µg/mL (Figure 4B).

Almost all the studied concentrations of the complex and the pure compound reduced the level of ROS in peritoneal macrophages (Figure 4C). The most pronounced effect was observed at the maximum studied concentrations and amounted to about 30% of the LPS-treated cells. Similar results were obtained for NO measurement (Figure 4D). The maximum effect was observed at high concentrations of the CRG/Ech complex (112 + 12.5 µg/mL) and 12.5 µg/mL of Ech alone, and amounted to 11% and 15%, respectively.

### 2.5. Protective Effect of Ech, CRG and CRG/Ech on Mice at LPS-Endotoximia

Ech, CRG, and CRG/Ech were delivered orally to mice once, at a dose of 100 mg/kg, 24 h before the induction of experimental endotoxemia driven by the intraperitoneal injection of LPS (*E. coli*) once, at a dose of 1 mg/kg.

Animal studies have shown that the introduction of LPS into intact mice causes significant deviations from the normal physiological masses of the liver, thymus and adrenal glands. The organo-somatic index (OSI) is an indicator of the negative effects of LPS and allows us to assess the degree of endotoxemia in mice. The OSI of the liver was higher than the control value by 21%, and that of the adrenal glands was 35% higher; however, the OSI of the thymus was 36% below the norm (Table 1). The size of the liver is largely determined by the intensity of its metabolism, so an increase in its OSI suggests a metabolic disorder in the organ. The administration of CRG, Ech, and CRG/Ech led to the minimization of disorders of physiological and metabolic homeostasis in mice subjected to LPS intoxication. This is indicated by the values of the OSI of the organs, which differed from the control by 4–15% (Table 1). The indicators of the OSI of the thymus and adrenal glands were closest to the control ones in the group of animals treated with the CRG/Ech complex.

*E. coli* LPS inhibited the process of energy supply in the body, which manifested itself in a decrease in the levels of adenosine triphosphate (ATP) and glycogen in the liver (Table 2). Additionally, clear acidosis in the liver tissue was noted. The lactate level was significantly higher than the control value. CRG, Ech, and the CRG/Ech complex supported bioenergetic processes in the body and largely prevented ATP deficiency, glycogenolysis, and acidosis in the liver. Thus, the contents of ATP and glycogen were lower than those in the control by 3–9% and 10–20%, while in the case of LPS, they were lower by 27% and 32%, respectively. With the CRG/Ech complex, the level of lactate in the liver tissue differed from that in the control by 11%. For CRG, the difference was 22%, and for Ech, 27% (compared to that for LPS without the drugs, 37%).

Thus, CRG, Ech, and their complex CRG/Ech minimize the pathophysiological manifestations of endotoxemia caused by *E. coli* LPS in mice. CRG and Ech showed highly similar protective activity under the conditions of bacterial endotoxemia. The CRG/Ech complex demonstrated the most effective neutralizing effect on LPS in terms of lactate and glycogen levels.

We examined the influence of CRG, Ech, and CRG/ECh on the in vivo production of anti-inflammatory and proinflammatory cytokines (IL-10 and TNF-α, respectively) in an experimental endotoxemia model induced by LPS. The IL-10 levels in the sera of the group of mice that received CRG/Ech were slightly higher than those in the controls, while the activity of Ech alone was at the level of the control (Appendix A). The application of the CRG/Ech complex and CRG, one day before the injection of LPS, promoted a 1.5–2-fold reduction in TNF-α production in mouse blood cells compared with the group of animals that did not receive samples before the induction of endotoxemia (Figure 5). Individual Ech had no effect on the decrease in the induction of the proinflammatory cytokine TNF-α.

## 3. Discussion

It is well-established that endotoxins, when entering into the bloodstream during local or systemic Gram-negative infections, impact almost all the systems of organisms, causing a number of pathophysiological changes. Due to the well-known resistance of organisms to antibiotics, it is important to search for neutralizing antibacterial drugs among natural substances. Previously, we showed the inhibitory effects of the marine sulfated polysaccharides CRGs on LPS-induced inflammation in a mouse model of endotoxemia and in the complex therapy of patients with enteric infections of Salmonella etiology. The preventive oral administration of CRGs significantly reduced the morphological, endocrine, and metabolic disorders caused by endotoxins [22]. In the present study, CRG was used as a matrix to contain Ech. Ech is known to have beneficial effects against various diseases. Through its antioxidant capacity, Ech is known to decrease LPS-induced reactive oxygen species (ROS) production in kidney epithelial cells [29]. In an animal model, Ech has been shown to reduce the intraocular inflammation caused by endotoxin-induced uveitis by reducing ROS production, as well as by reducing the expression of NF-kB and TNF-α [26]. EchA administration has been shown to protect the liver from sepsis-induced liver damage by counteracting hepatic oxidative stress. The antioxidant effects of EchA successfully reduced lipid peroxidation and tissue damage in liver tissue in the puncture (CLP) sepsis rat model [30].

The main problem with Ech is its low solubility and oxidative degradation. The results that we obtained earlier showed that CRGs improve the solubility of Ech and prevent its oxidation [25].

Molecular docking was used to study the interaction of CRGs with Ech. In this work, by computer simulation, we have shown that Ech interacts with carrageenan and forms a complex with it. H-donor and H-acceptor bonds are involved in the formation of the complex. This confirms our earlier spectroscopy data on the formation of the CRG/Ech complex [25].

One of the objectives of this work was to study the anti-inflammatory effects of the CRG/Ech complex using an in vitro model. It was shown that the marine sulfated polysaccharide has a protective effect and significantly reduces the secretion of proinflammatory cytokines, as well as NO in mouse macrophages activated by LPS [31]. Macrophages are important innate immune cells that are responsible for the initiation phase of inflammation [32]. LPS is a component of the cell wall in Gram-negative bacteria, which stimulates inflammatory responses in macrophages. Thus, in the present study, LPS-stimulated murine macrophage cells (RAW 264.7) and peritoneal macrophages isolated from the peritoneal cavity in the mouse Balb/c line were used as a model to investigate the anti-inflammatory activity of the CRG/Ech complex. It is known that the induction of inflammatory cytokines is dependent on the upregulation of ROS [33]. Substances, such as resveratrol and astaxanthin, have been reported to ameliorate endotoxin-induced uveitis by suppressing ROS production [34,35]. Increased ROS production is one of the factors that triggers inflammation. Therefore, reducing ROS levels under LPS exposure in cells may be beneficial in limiting the inflammatory response.

The study of the action of LPS and samples on the production of ROS and NO in the murine macrophage cell line RAW 264.7 showed that the combined action of LPS (1 µg/mL) with Ech and the CRG/Ech complex resulted in a marked decrease in the synthesis of ROS and NO. The CRG/Ech complex, after incubation with cells for 24 h, decreased the ROS formation induced by LPS by approximately 50%, regardless of the concentration of Ech. However, the production of NO was remarkably and concentration-dependently reduced in CRG/Ech-treated RAW 264.7 cells.

Peritoneal macrophages also upregulated ROS upon stimulation with LPS, while CRG/Ech and Ech inhibited the LPS-induced activation of ROS, especially at a high sample concentration. The influence of Ech and CRGs on the activation of NO was significant regardless of the concentration, and their action was comparable with that of the negative control.

In the present study, we examined the effect of the CRG/Ech complex and its components on LPS-induced endotoxemia in mice by the degree of the variability of biochemical and pathological parameters that are relevant to the formation of adequate responses to stressors, including bacterial endotoxins. In our experiments, LPS, parenterally injected into mice (at a nonlethal dose), caused significant changes in the organism’s physiological status. This was indicated by changes in the organo-somatic indices, as shown in Table 1. It was found that *E. coli* LPS injection caused considerable changes in the masses of the liver, thymus and adrenal glands compared to the physiological norms. The size of the liver is largely determined by the intensity of the metabolism in it; therefore, an increase in its OSI implies a disruption of metabolism in the organ. Thymus involution and adrenal hypertrophy indicate pronounced stress responses in mice caused by endotoxemia. The results of this study show that the administration of the CRG/Ech complex and its components, CRG and Ech, led to the minimization of disorders of physiological and metabolic processes in mice subjected to LPS intoxication. The indicators of the OSIs of the thymus and adrenal glands were closest to the control values in the group of animals treated with the CRG/Ech complex. In the pathogenesis of LPS-induced endotoxemia, disorders of energy metabolism play a leading role. The application of CRG, Ech, and the CRG/Ech complex optimized bioenergetic processes in the body: it prevented pathogenic ATP deficiency, glycogenolysis, and acidosis in the liver. The results of our study have shown that the endotoxemia induced by the intraperitoneal injection of bacterial LPS was less pronounced in the mice that received CRGs, Ech, and the complex. The CRG/Ech complex minimized the pathophysiological manifestations of endotoxemia caused by *E. coli* LPS in mice. CRG and Ech showed practically similar protective activity under conditions of bacterial endotoxemia. The most effective neutralizing effect on LPS in terms of lactate and glycogen levels was demonstrated by the CRG/Ech complex.

We revealed the protective action of the CRG/Ech complex during endotoxemia, exhibited by a decrease in TNF-α levels in the mouse serum. It is known that the concentration of TNF-α in the blood increases rapidly during endotoxemia. The production of IL-10 inhibits the synthesis of TNF-α. It should be noted that the CRG/Ech complex administrated orally and CRG alone decreased TNF-α synthesis compared with controls. At the same time, the CRG/Ech complex had no similar effect on the induction of the anti-inflammatory cytokine IL-10. As shown, the cytokine levels in the sera of mice treated with the CRG/Ech complex and CRG were comparable to those in the control group.

## 4. Materials and Methods

### 4.1. Polysaccharide

Red algae *Chondrus armatus* were harvested at the Peter the Great Bay, Sea of Japan and identified on the morphological and anatomical characteristics by Prof. E. Titlynov and T. Titlynova (*National Scientific Center of Marine Biology*, *Far-Eastern Branch of the Russian Academy of Sciences*) using an electron microscope. Dried and milled algae (50 g) were suspended in hot water (1.5 L) and the polysaccharides were extracted three times at 80 °C for 3 h in a water bath. Three hot extracts were combined, centrifuged at 4000 rpm^−1^ to remove residues of the cell wall, filtered through a Vivaflow200 membrane (Sartorius, Gottingen, Germany) with a pore size of 100 kDa, concentrated on a rotary evaporator, and precipitated polysaccharides with a triple volume of 96% ethanol. The resulting precipitate was dissolved in water and fractionated with potassium salt into KCl-insoluble and non-gelling KC1-soluble fractions as described previously and their structures were established according to the published protocol [28]. Gelling KCl-insoluble fraction as κ-carrageenan (κ-CRG) was used.

Commercial LPS from *Escherichia coli* 055:B5 (Cat No: L2880, Lot No: 102M4017V, Sigma, St. Louis, MO, USA) was used in the study.

### 4.2. Preparation of Ech Water Solutions with CRG (CRG/Ech)

The substance Echinochrome (registration number in the Russian Federation is P N002362/01 (State Register of Drugs (as of 5 December 2016) Part 2)) was obtained in G.B. Elyakov Pacific Institute of Bioorganic Chemistry in powder form. Ethanolic solution of Ech in concentration 10 mg/mL was used as a stock solution. A solution of 1% CRG was prepared by dissolving CRG in deionized water at 50 °C. Hence, 0.1 mL of a stock solution of Ech was added to 10 mL of 1% CRG water solutions at 25 °C by mixing. As a result, Ech water solutions with CRG were obtained. The concentration of Ech in the solution was measured using absorption spectra at λ = 468 nm.

### 4.3. Cell Culture

The murine macrophages cell line RAW264.7 cells were purchased from ATTC (TIB-71™; American Type Culture Collection, Manassas, VA, USA). RAW264.7 cells were culture in DMEM with 10% heat-inactivated fetal bovine serum, 100 U/mL penicillin and 100 μg/mL streptomycin. Cells were maintained in humidified atmosphere at 37 °C with 5% CO_2_.

Peritoneal macrophages were isolated from mouse Balb/c line peritoneal cavity. After isolation, peritoneal macrophages were seeded in a 10 cm petri dish for 2 h. Then, the peritoneal macrophages were washed 2 times with PBS, counted, and resuspended in DMEM with 10% heat-inactivated fetal bovine serum, 100 U/mL penicillin, and 100 μg/mL streptomycin.

### 4.4. Cell Viability Assay (MTT Test)

RAW 264.7 cells (2 × 10^4^ cells/well) were incubated in a CO_2_ incubator for 24 h at 37 °C for adhesion. After that 20 μL of test solution were loaded to the cells and incubated 24 h. After incubation, the medium with tested substances was replaced by 100 μL of fresh medium. Then, 10 μL of MTT (3-(4,5-dimethylthiazol-2-yl)-2,5- diphenyltetrazolium bromide) (Sigma-Aldrich, St. Louis, MO, USA) stock solution (5 mg/mL) was added to each well and the microplate was incubated for 4 h. After that, 100 μL of SDS-HCl solution (1 g SDS/10 mL dH_2_O/17 μL 6 N HCl) was added to each well, followed by incubation for 18 h. The absorbance of the converted dye formazan was measured using a Multiskan FC microplate photometer (Thermo Scientific, St. Louis, MO, USA) at a wavelength of 570 nm. All experiments were repeated in triplicate. Cytotoxic activity was expressed as the percent of cell viability.

### 4.5. ROS and NO Level Analysis

After 24 h of adhesion, RAW 264.7 cells (2 × 10^4^ cells/well) or peritoneal macrophages were incubated with compounds (Ech, CRG and CRG/Ech) at different concentrations for 1 h. Then, LPS at a concentration of 1 μg/mL was added to each well, and cells were incubated for 1 or 24 h. To study ROS formation, 20 µL of 2,7-dichlorodihydrofluorescein diacetate solution (10 µM H2DCF-DA, Molecular Probes, St. Louis, MO, USA) was added to each well, such that the final concentration was 10 µM, and the microplate was incubated for an additional 10 min at 37 °C in the dark.

In the case of determining the NO production, a DAF-FM fluorescent probe solution (5 μM, Molecular Probes, MO, USA) was added to each well, and the microplate was further incubated for 40 min at 37 °C in the dark. In both cases, the fluorescence intensity was measured using a high-speed plate reader PHERAstar FS (BMG Labtech, Ortenberg, Germany) at λex = 485 nm and λem = 518 nm. The data were processed by MARS Data Analysis v. 3.01R2 (BMG Labtech, Ortenberg, Germany). The results were presented as a percentage of positive control data. Statistical analysis was performed using Student’s t-criterion for unpaired data, and *p*-values of less than 0.05 were considered significant. All data presented as mean ± standard deviation.

### 4.6. Animal Experiments

The work was carried out in accordance with “Directives 2010/63/EU of the European Parliament and the Council of the European Union for the Protection of Animals used for Scientific Goals”. The animal study was approved by Ethical Committee of G.B. Elyakov Pacific Institute of Bioorganic Chemistry, Far Eastern Branch, Russian Academy of Sciences: Protocol code 06/20 from 25 December 2020.

Mature male CD-1 mice were obtained from the G.B. Elyakov Pacific Institute of Bioorganic Chemistry, FEB RAS (Vladivostok, Russia). The maintenance and euthanasia of animals were correspondent to the principles of Directive 2010/63/EU of the European Parliament and of the Council on the Protection of Animals Used for Scientific Purposes.

The mice with body mass of 28–32 g were kept in standard conditions of the vivarium at a controlled temperature 20–22 °C and ambient humidity 60–65%. Light was maintained on an artificial 12 h light-dark cycle. Each experimental group consisted of six animals, each mouse in the cage had a floor area of 70 cm^2^, which corresponds to international standards. The mice provided with water standard feed compliant GOST R 50258-92 (CJSC ProKorm, Russia) ad libitum. The mice were randomly allocated into five groups: control (NaCl) group, 1, LPS-group, and 3 experimental groups (CRG, Ech, CRG/Ech). The control group received intraperitoneally 0.5 mL saline. All animals except the control group were given intraperitoneally LPS solution in dose 1 mg/kg body mass (0.1 mg/mL, pH 7.0). The mice of “control” and “LPS” groups were given only standard feed whereas mice of other groups daily administrated Ech, CRG, and CRG/Ech suspensions in dose 100 mg/kg body mass (4 mg/mL, pH 7.0) 24 h before LPS injection through gastric gavage. At the end of experiment mice were killed by decapitation, inner organ was removed, weighed. Determination of organo-somatic indexes (OSI) expresses the weight of each organ as a percentage of body weight with the following formula: W of the organ (g)/W of mouse (g) × 100 [36]. The glycogen content in liver was estimated with the anthrone reagent [37]. The adenosine triphosphate (ATP) and lactate levels in the liver were measured using enzymatic pectrophotometric methods with a test-system comprising nicotinamide coenzymes NADF and NADH, respectively [38,39].

Blood was collected in tubes by decollation mice of groups 1 and 3 24 h after LPS injection. Blood was centrifuged for 5 min at 400 g, and serum was collected and stored frozen at −22 °C until it was assayed regarding the content of pro-inflammatory and anti-inflammatory cytokines.

#### 4.6.1. Cytokine Assays

Sandwich enzyme-linked immunosorbent assays (ELISA) were performed to quantify tumor necrosis factor a (TNF-α) and interleukin-10 (IL-10) in the serum according to the manufacturer’s instructions using affinity purified monoclonal anti-mouse TNF-α and IL-10 (eBioscience, St. Louis, MO, USA) as the capture antibodies, and biotin-conjugated TNF-α and IL-10 (eBioscience, USA) as the detection antibodies, respectively. HRP-labeled avidin (eBioscience) was used for detection. Cytokine concentrations were calculated from the absorbance values by plotting the values against the TNF-α and IL-10 standard curves, which was performed for each assay. The sensitivity of the ELISA assays was 2.9 pg/mL for TNF-α.

#### 4.6.2. Statistical Analysis

All measurements were performed in three replicates. All results were expressed as mean ± the standard deviation compared by ANOVA; all difference were considered to be statistically significant if *p* < 0.05. Data were analyzed using the software Statistic 6.0. (StastSoft, Jefferson, MO, USA). To confirm the normal distribution of variables was investigated using Shapiro-Wilks test.

### 4.7. Computer Modelling

The model of the κ-CRG double helix fragment was obtained using the crystal structure of iota-carrageenan (PDB ID 1CAR) (www.wwpdb.org; accessed on 1 January 2020 H.M. Berman, K. Henrick, H. Nakamura (2003) Announcing the worldwide Protein Data Bank Nature Structural Biology 10 (12): 980; Arnott S, Scott WE, Rees DA, McNab CG) [40] and replacing the sulfate group in the 3,6-anhydro-alpha-D-galactopyranosyl with hydrogen using the program Molecular Operating Environment (MOE, Chemical Computing Group ULC; Montreal, QC, Canada: September 2020). The κ-CRG model was solvated, optimized using the Amber:EHT10 force potential, and used for molecular docking with Ech. The structure of Ech was obtained from the Cambridge X-ray Diffraction Database (CCDC ID NERLUS) (Groom C.R., Bruno I.J., Lightfoot M.P., Ward S.C. The Cambridge Structural Database. Acta Cryst. 2016; B72:171–179. https://doi.org/10.1107/S2052520616003954) [41]. Molecular docking was performed using the geometric docking program Global Range Molecular Matching GRAMM v1.03 [42] with high resolution docking parameters (Grid step 1.7 Å). The analysis of contacts of the κ-CRG complex with Ech with the lowest energy was performed using the ligand interaction module of the MOE program. The results were obtained using the equipment of the Shared Resource Center, Far Eastern Computing Resource of the Institute of Automation and Control Processes Far Eastern Branch of the Russian Academy of Sciences (IACP FEB RAS, available online: https://cc.dvo.ru, accessed on 22 August 2022).

## 5. Conclusions

In the present study, we evaluated the protective effect of marine substances CRG and Ech in experimental endotoxemia. In this work, we proved by computer simulations that CRG forms a complex with Ech. The anti-inflammatory effects of the CRG/Ech complex were investigated in vivo in LPS-induced endotoxemia in mice and in vitro in LPS-stimulated RAW 264.7 cells and peritoneal macrophages. The results demonstrated that the CRG/Ech complex suppressed the LPS-induced inflammatory response by reducing the production of ROS and NO in the macrophages. Furthermore, the in vivo experiment indicated that the CRG/Ech complex inhibited disorders of physiological and metabolic homeostasis in mice exposed to LPS intoxication and suppressed the dysregulation of and reduced the levels of proinflammatory cytokines in the mouse serum.

These results demonstrate that the CRG/Ech complex possesses potent in vitro and in vivo anti-inflammatory effects and can be useful for both oral and topical administration for reducing endotoxemia under the development of the infectious process.

## Figures and Tables

**Figure 1 ijms-23-11702-f001:**
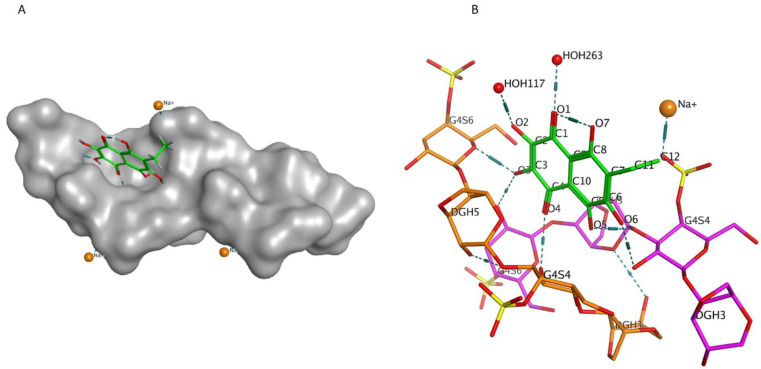
(**A**)—Model of the Ech with the κ-CRG double helix fragment. The molecular surface of CRG is shown in grey. The structure of Ech is shown as stick in green. (**B**)—Ech binding site with κ-CRG. The structure of κ-CRG double helix residues is shown in orange and pink, Ech is shown as stick in green, water molecules and sodium ion are shown as specific in red and yellow. Hydrogen bonds are shown in dotted lines.

**Figure 2 ijms-23-11702-f002:**
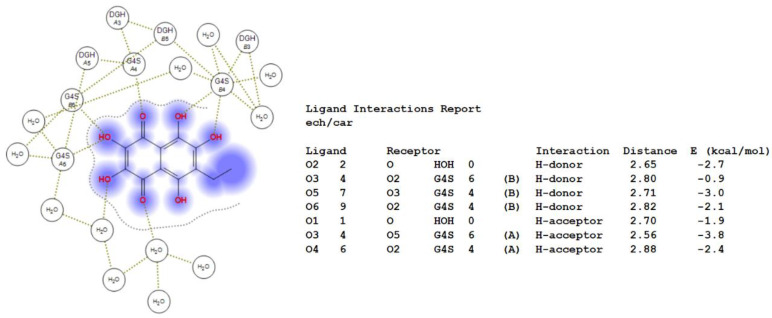
2D-diagram of contacts of Ech with the κ-CRG in the binding site and table of Ech interaction contacts.

**Figure 3 ijms-23-11702-f003:**
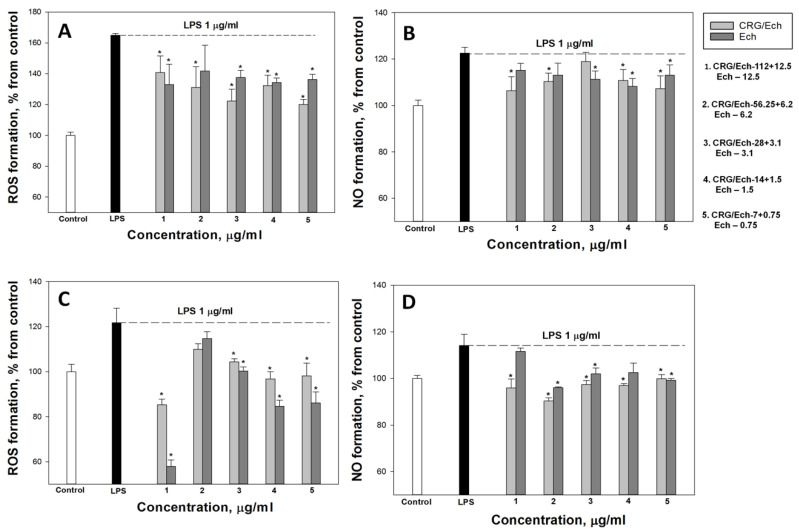
Effects of the CRG/Ech or Ech on intracellular ROS and NO formation in the murine macrophages cell line RAW 264.7 and peritoneal macrophages. The effects of the CRG/Ech or Ech on ROS formation into RAW264.7cells (**A**) and peritoneal macrophages (**C**) treated by 1 µg/mL LPS. The effects of the CRG/Ech or Ech on NO formation into RAW 264.7cells (**B**) and peritoneal macrophages (**D**) treated by 1 µg/mL LPS. Cells were incubated with CRG/Ech or Ech for 1 h at 37 °C, then 24 h with LPS. The data are shown as the means ± SE; *—*p* < 0.05.

**Figure 4 ijms-23-11702-f004:**
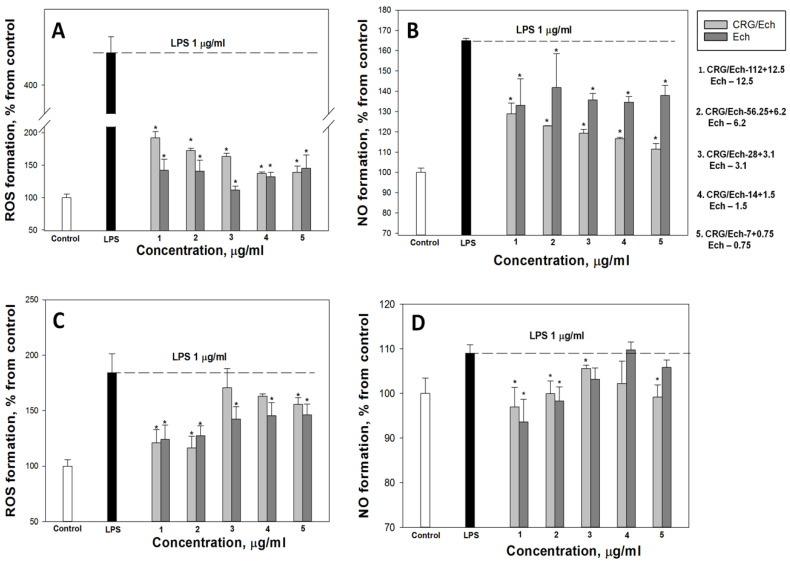
Effects of the CRG/Ech or Ech on intracellular ROS and NO formation in the murine macrophages cell line RAW 264.7 and peritoneal macrophages. The effects of the CRG/Ech or Ech on ROS formation into RAW264.7 cells (**A**) and peritoneal macrophages (**C**) treated by 1 µg/mL LPS. The effects of the CRG/Ech or Ech on NO formation into RAW 264.7 cells (**B**) and peritoneal macrophages (**D**) treated by 1 µg/mL LPS. Cells were incubated with CRG/Ech or Ech for 24 h at 37 °C, then 24 h with LPS. The data are shown as the means ± SE; *—*p* < 0.05.

**Figure 5 ijms-23-11702-f005:**
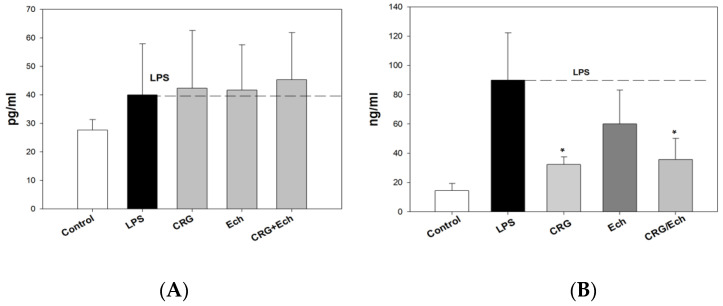
The levels of anti-inflammatory Il-10 (**A**) and pro-inflammatory TNF-α (**B**) cytokines in mice sera under oral administration of—CRG, Ech, CRG/Ech (24 h before injection of LPS). The dose of orally sample was 100 mg/kg. Values are means 6 SD (n 5 10). The data are shown as the means ± SE; * Significantly different from control (saline), *p* < 0.05.

**Table 1 ijms-23-11702-t001:** The effect of CRG, Ech and CRG/Ech on some physiological parameters in mice intoxicated with *E. coli* LPS.

Groups	Organo-Somatic Index, %		
	Liver	Thymus	Adrenals
control	4.81 ± 0.206	0.176 ± 0.011	0.023 ± 0.001
LPS	5.80 ± 0.306 *	0.112 ± 0.008 *	0.031 ± 0.002 *
LPS + CRG/Ech	5.01 ± 0.208	0.163 ± 0.007 **	0.025 ± 0.001 **
LPS + CRG	4.92 ± 0.230 **	0.149 ± 0.006 **	0.026 ± 0.001 **
LPS + Ech	4.91 ± 0.225 **	0.155 ± 0.005 **	0.026 ± 0.001 **

LPS—*E. coli* lipopolysaccharide; CRG—carrageenan; Ech—echinochrome; CRG/Ech—complex (10:1); * mean ± SEM (n = 6 observations in “control” group, 7 observation in other groups);—*p* < 0.05—compared with «control», **—*p* < 0.05—compared with LPS used Student *t*-test.

**Table 2 ijms-23-11702-t002:** The effect of the CRG, Ech and CRG/Ech complex on the content of energy metabolites in the liver of mice at LPS- endotoxemia.

Groups	Metabolits, μmol/g		
	ATP	Glycogen	Lactate
control	2.68 ± 0.13	220.7 ± 16.8	1.92 ± 0.12
LPS	1.96 ± 0.11 *	150.8 ± 12.5 *	2.63 ± 0.14 *
LPS + CRG/Ech	2.60 ± 0.11 **	198.7 ± 13.3 **	2.14 ± 0.10 **
LPS + CRG	2.43 ± 0.12 **	180.4 ± 12.6	2.35 ± 0.10
LPS + Ech	2.50 ± 0.14 **	175.5 ± 10.3	2.44 ± 0.16

ATP—adenosine triphosphate, *—*p* < 0.05—compared with “control” group, ** *p* < 0.05 compared with “LPS” group used Student’s *t*-test.

## Data Availability

Not applicable.

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
