# Peer review of "The Anti-Inflammatory Effect of Carrageenan/Echinochrom Complex at Experimental Endotoxemia"

_ijms, 2022, doi:10.3390/ijms231911702_

Round 1
Reviewer 1 Report
The authors would like to show the neutralization properties of carrageenan/echinochrome after LPS endotoxemia. It is an interesting approach, however some suggestions have to be made.
1. The effect of LPS alone seems very low in RAW cells. Here I would suggest to take another cell line, which shows high response to LPS. I would suggest a human cell lime (THP-1,A549) or primary human cells ( e.g. monocytes). In addition, as the effect of LPS is always time-depended please show a time course of LPS stimulation. Furthermore beside NO und ROS production, cytokine release in RAW cells would be interesting.
2. Unfortunately figure 1 has bad quality, so I cannot read the amount of substances in RAW cells. Again, was a time series performed with the two substances?
3. Experimental design: Why do you pretreat the mice? It is always closer to the clinical setting to perform a “curative” treatment with the “drugs” after onset of endotoxemia.
4. Please define the OSI in the methods: This has to be clarified. Otherwise I cannot judge the performance of this experiments.
Minor points:
- Grammer: gramnegative line 26
- Line 48 problem removal?
- Line 63 it was shown
- In vivo and in vitro in italics
- Introduction is too long, please come to point more quickly
- Figures are of bad quality
- Please write a conclusion at the end
- Please show Cytokine production in mice, were data not shown is indicated
- English has to be proven extensively: However as I am not a native speaker I feel not qualified to write this in the evaluation table.
Author Response
Rew1
We are grateful to Reviewer for a thorough analysis of our manuscript. We have made changes to the article in accordance with your comments and present our answer .
The authors would like to show the neutralization properties of carrageenan/echinochrome after LPS endotoxemia. It is an interesting approach, however some suggestions have to be made.
- The effect of LPS alone seems very low in RAW cells. Here I would suggest to take another cell line, which shows high response to LPS. I would suggest a human cell lime (THP-1,A549) or primary human cells ( e.g. monocytes). In addition, as the effect of LPS is always time-depended please show a time course of LPS stimulation. Furthermore beside NO und ROS production, cytokine release in RAW cells would be interesting.
Answer
- a) We do not consider the effect of LPS alone to be very low. In the experiment, we used of LPS 1 µg/ml. At this concentration, LPS increased ROS synthesis by more than 2 times compared with the control. A similar value of the inducing activity of LPS on ROS on the RAW cells was obtained by other authors (J Food Biochem. 2019;43:e12994. ileyonlinelibrary.com/journal/jfbc and , Antioxidants 2021, 10, 1208 https://doi.org/10.1111/jfbc.12994 ) and also by us on Human blood cells
- b) We have previously investigated the effect of CRG on the biological properties of Ech using Human blood cells (Marine drug 2018, Sokolova et al). We showed that Ech reduced production ROS in neutrophils caused by CRG. We also showed that the inclusion of Ech in the CRGs decreases the ability of Ech to induce the expression of pro-inflammatory cytokines, especially TNFα, and increases the induction of anti-inflammatory cytokine IL-10. In the presented article we have shown that CRG forms a complex with Ech and this complex has a protective effect in LPS-induced endotoxemia. It is know that LPS is the primary target for the interaction with macrophages and the important innate immune cells that are responsible for the initiation phase of inflammation. That's why we used as the model to investigate the anti-inflammatory activity of CRG/Ech complex LPS-stimulated murine macrophages cell line RAW 264.7 and peritoneal macrophages. c).
с) We agree with you that the effect of LPS is time- dependent. In this work, we determined the action of LPS within 1 hour and 24 hours. We apologize that not all results were presented. We have completed the figure 3 and 4
- Unfortunately figure 1 has bad quality, so I cannot read the amount of substances in RAW cells. Again, was a time series performed with the two substances?
Answer
We apologize for the poor quality of the drawing. The Figure and its captions legends have been corrected. Exposure time dependency added ( Fig3 and 4)
- Experimental design: Why do you pretreat the mice? It is always closer to the clinical setting to perform a “curative” treatment with the “drugs” after onset of endotoxemia.
Answer:
Preventive the use of exogenous biologically active samples, in particular carrageenan and echinochrome, is advisable in the presence of risk factors for the development of presymptomatic diseases. Besides the design of this animal experiment was chosen based on the common use of carrageenan, which is the main component of the complex. As you know, carrageenan is used as an ingredient in many food products.
- Please define the OSI in the methods: This has to be clarified. Otherwise I cannot judge the performance of this experiments.
Answer:
The definition of OSI is presented in the methods
we supplemented the graphs in our work.
Minor points:
- Grammer: gramnegative line 26
Corrected
- Line 48 problem removal?
Corrected
- Line 63 it was shown
Corrected
- In vivo and in vitro in italics
Corrected
- Introduction is too long, please come to point more quickly
Introduction has been shortened
- Figures are of bad quality
Figures are corrected
- Please write a conclusion at the end
Conclusion has been added
- Please show Cytokine production in mice, were data not shown is indicated
Figure with Cytokine production in mice ( IL-10) was added
- English has to be proven extensively: However as I am not a native speaker I feel not qualified to write this in the evaluation table.
- English has been corrected by MDPI for English editing
Reviewer 2 Report
The manuscript "Anti-inflammatory effect of the carrageenan/echinochrome complex in experimental endotoxemia" is of interest to scientists working in the field of natural products. The article is a continuation of the work published earlier by Yermak I.M. et al. Carrageenans are sulfated polysaccharides from red algae as matrices for echinochrome incorporation. Mar. Drugs 2017, 15, 337. https://doi.org/10.3390/md15110337.
It is presented well but some information should be added and some typos should be corrected.
- Please unify "RAW 264.7" - sometimes with a space, sometimes without a space.
-Please unify text font - lines 88-89
-In addition, the Conclusion part is missing.
Perhaps a figure with the chemical structureы of resveratrol, hirsutanol A, and echinochrome (Ech) should be added to better understand the "structurу-activity" relationship.
Also, the cytotoxic activity data of CRG/Ech ( line 140) may be presented in supplementary materials.
Author Response
- Rew 2
The manuscript "Anti-inflammatory effect of the carrageenan/echinochrome complex in experimental endotoxemia" is of interest to scientists working in the field of natural products. The article is a continuation of the work published earlier by Yermak I.M. et al. Carrageenans are sulfated polysaccharides from red algae as matrices for echinochrome incorporation. Mar. Drugs 2017, 15, 337. https://doi.org/10.3390/md15110337.
It is presented well but some information should be added and some typos should be corrected.
Answer
We are grateful to Reviewer for the analysis of our manuscript. We have made changes to the article in accordance with your comments
- Please unify "RAW 264.7" - sometimes with a space, sometimes without a space.
corrected
-Please unify text font - lines 88-89
corrected
-In addition, the Conclusion part is missing.
Conclusion has been added
Perhaps a figure with the chemical structures of resveratrol, hirsutanol A, and echinochrome (Ech) should be added to better understand the "structurу-activity" relationship.
The structure of Ech and CRG were added in the supplementary
Also, the cytotoxic activity data of CRG/Ech ( line 140) may be presented in supplementary materials.
Figure with the cytotoxic activity data of CRG/Ech was added in supplementary
Round 2
Reviewer 1 Report
Thank you very much for improoving your manuscript!